# Determinants of rice production and market supply: A study of Bench Sheko zone in Ethiopia

**Ejigu Mulatu** 👤 *

Policy Studies Institute, Addis Ababa, Ethiopia

* ejeta2010@gmail.com

**Data Availability Statement:** All relevant data are within the manuscript and its Supporting Information files.

## Abstract

In Ethiopia rice crop is considered as a strategic food security crop which is expected to contribute to ensuring food security in the country. Bennch Sheko Zone is one of the major rice growing areas in the South Western Regional State. The study was conducted with specific objectives to investigate factors affecting smallholder farmers' market supply of rice and identify constraints related to rice production in the study area. Two-stage sampling technique was employed to select 119 representative rice-producer households. Descriptive statistics and appropriate econometric models were used to analyze the collected data. Multiple linear regression model used to analyze factors affecting rice market supply. Descriptive result of the study showed, the average annual rice production at the household level was 2.8 tons, of which 70% was supplied to the market. Econometric result showed farm size owned, credit use, annual income, number of oxen owned, and quantity of rice produced were found to be significantly affecting the market supply of rice in a district. Major constraints related to rice production in the district were a lack of proper weed management practices, improved seed, proper method and time of fertilizer application, weak institutional support, disease, and post-harvest handling problems were also important. The research findings suggest that attention should be given to rice production constraints through generation and wide demonstration of demand-driven rice production and post-harvest handling technologies for increased production and productivity to have a better market supply of rice to the market and benefit smallholder farmers.

## 1. Introduction

Rice is the most significant food crop in the world and it has been considered to assist as a chief food basis for more than 50% of the world's population for the long years [1]. Globally, annual rice production has risen to 758.8 million tons of paddy rice. In addition to its contribution to consumption, it is a good opportunity for domestic and international markets for economic development mainly in China, India Indonesia, and USA, and other countries [2,3]. These factors have drawn a lot of interest from various stakeholders, and as a result, its production has spread to many parts of the globe.

**Funding:** The author(s) received no specific funding for this work.

**Competing interests:** The author have declared that no competing interests exist.

Although rice was introduced to Ethiopia very recently, rice production revealed increasing trend in area and productivity. Following the increasing demand of rice and its production potential, rice is classified as a fourth "National Food Security Crop" after wheat, maize, and tef [4]. In short time period it has proven to be a crop that can assure food and livelihood security [5]. Currently, rice in Ethiopia is considered as a strategic food security crop that has received due emphasis in the promotion of agricultural production, and as such it is considered as the "millennium crop" expected to support in ensuring food security in a country.

Improving rice varieties through research and development was one of the sector's main priorities. Over the past 20 years, a number of rice varieties that have showed production increment of 44.1% (6.8 tons/ha) at the research level have been released from research institutions [6]. The lack of mechanization, grain quality characteristic, extension service, and poor rice utilization, however, makes only a small number of them adopted by rice producers. As a result, despite productivity gains at the research level, mean national productivity (2.8 tone/ha) remained relatively low in comparison to the global average productivity (4.4 tones/ha) [7].

It is argued that rice remains a minor crop regarding area coverage and production as compared to potential large areas and favorable agroclimatic conditions for expanding rice production. A study by [8] showed major drivers of rice yield among producers. Accordingly, water stress, low soil fertility, lack of draft animals, shortage of credit, pests, weak extension, weeds, labor shortage, increase input price, flooding, poor marketing, and the lack of storage, poor seed system, post-harvest losses, lack of farm tools, and price fluctuation [8].

Constraints for sustainable production and productivity of rice as mentioned by various studies include poor access to improved rice varieties; poor access and use of modern post-harvest techniques and equipment; grassy weeds and insect pests; limited access to credit; shortage of labor; poor knowledge of producers and other market actors about rice product quality; excessive numbers of intermediaries, and price seasonality; and inadequate storage facilities [9–11]. Even though industrial fertilizer application increases rice yield, the lack of effective fertilizer application affected the production and productivity of rice [12]. On the other hand, demographic and socio-economic factors including household labor availability, land holding, distance to the nearest village market, access to agricultural extension, access to the source of rice seeds, and access to new cultivars of rice affected rice technology adoption, production and output marketing [11,13].

The two main problems influencing the rice sector's inability to compete effectively with imported rice and penetrate into both domestic and international markets are a lack of mechanization in the production of rice and low-quality rice grains [14]. In addition to production related constraints, different factors affect the market supply of rice. Education level and quantity of rice produced affect the volume of rice sales positively but family size determines the volume of sales negatively and quantity produced jointly affected both the probability of market participation and volume of supply positively. The gender of the household head, access to improved seeds, years of formal education, and average rice yield influenced the probability that a farmer would participate in the market [11,15].

The Bench Sheko zone in general and the Guraferda area in particular are important rice-growing areas in the southwest of Ethiopia among the key producing regions [15]. There were various previous studies conducted on rice in the study area. According to a [15,16] study, rice is one of the major cash crops in Guraferda District as all producer farmers supplied rice produce to the market in the production season. Their study mainly focused on marketing aspects giving little focus on production-related constraints.

The [17] study assessed the economic efficiency of smallholder farmers in rice production in the district. However, their findings solely focused on measuring efficiency level and its determinants, showing little about production status and marketing-related problems. A study

by [18] on production expansion and comparative advantage of upland rice production and its effect on the local farming systems showed better production status of rice in the district. However, rice marketing-related factors have not been assessed intensively. This study would add value by assessing production and marketing-related issues together. The study aims to come up with evidences targeted for policy suggestions. The research findings are expected to provide evidences for governmental and non-government institutions in supporting rice producer farmers through effective intervention on significant determinants of rice production and market supply. Therefore, this study was conducted with specific objectives to identify factors affecting smallholder farmers' market supply of rice and to assess constraints related to rice production and marketing in the district.

## 2. Methodology

### 2.1 Description of the study area

The study was undertaken in a major rice-growing area of the Guraferda district in the Bench Sheko zone of South West Ethiopian Region. Guraferda district is located at longitude 34088' - 36014'E and latitude 50033'-70.21'N [16]. It has a population of about 43,253 persons, of which 23,473 are males and 19,664 are females. It is bordered on the south by the Bero district, on the west and north by the Gambela Region, on the northeast by Sheko, on the east by South Bench, and on the southeast by Menit Shasha (shown in Fig 1). The estimated area of Guraferda district is about 2565.42 km2. There are 27 kebeles in the district. Based on the data obtained from the Agricultural and Rural Development Office of Guraferda, the population of the district was expected to exceed 45,028 [17]. In the district, agriculture forms the major livelihood as crops grown were rice, sorghum, maize, sesame, haricot bean, coffee, ground nut, spices like pepper, fruits such as orange, mango, papaya, and vegetables like banana, potato, cabbages, etc. Rice, sorghum, maize, and coffee were grown to a larger extent [17]. The altitude of the district ranges from 814 to 1995 meters above sea level. The temperature ranges between 10˚c to 30˚c. The mean annual rainfall is 1710 millimeters [16].

### 2.2 Sampling techniques and sample size determination

Two-stage sampling strategy was employed. Primarily, four sample rice producing *Kebeles* were selected randomly as they represent the whole district. In the second stage, by taking the list of rice producer household heads in each kebele, representative sample size was determined using the formula which was developed by Yemane cited in [18]: $n = \frac{N}{1+N(e)^2}$ where, n is sample size; N is target population, and e is level of precision. Based on this formula, by assuming level of precision 9%, and given number of total rice producer households in the district, sample size was estimated to be about 119.

$$n = \frac{3296}{1 + 3296(0.09)^2} = 119$$

### 2.3 Data Sources and methods of collection

In this study, data has been collected from both primary and secondary data sources. Primary data mainly collected regarding the demographic and socio-economic profile of rice producers. A structured questionnaire was used to generate the primary data from the selected sample of smallholder rice producers. The primary data were collected from the sample respondents and focus group discussions (FGDs) and key informants' interviews were also used to gather necessary information to supplement data collected from selected respondents. Secondary

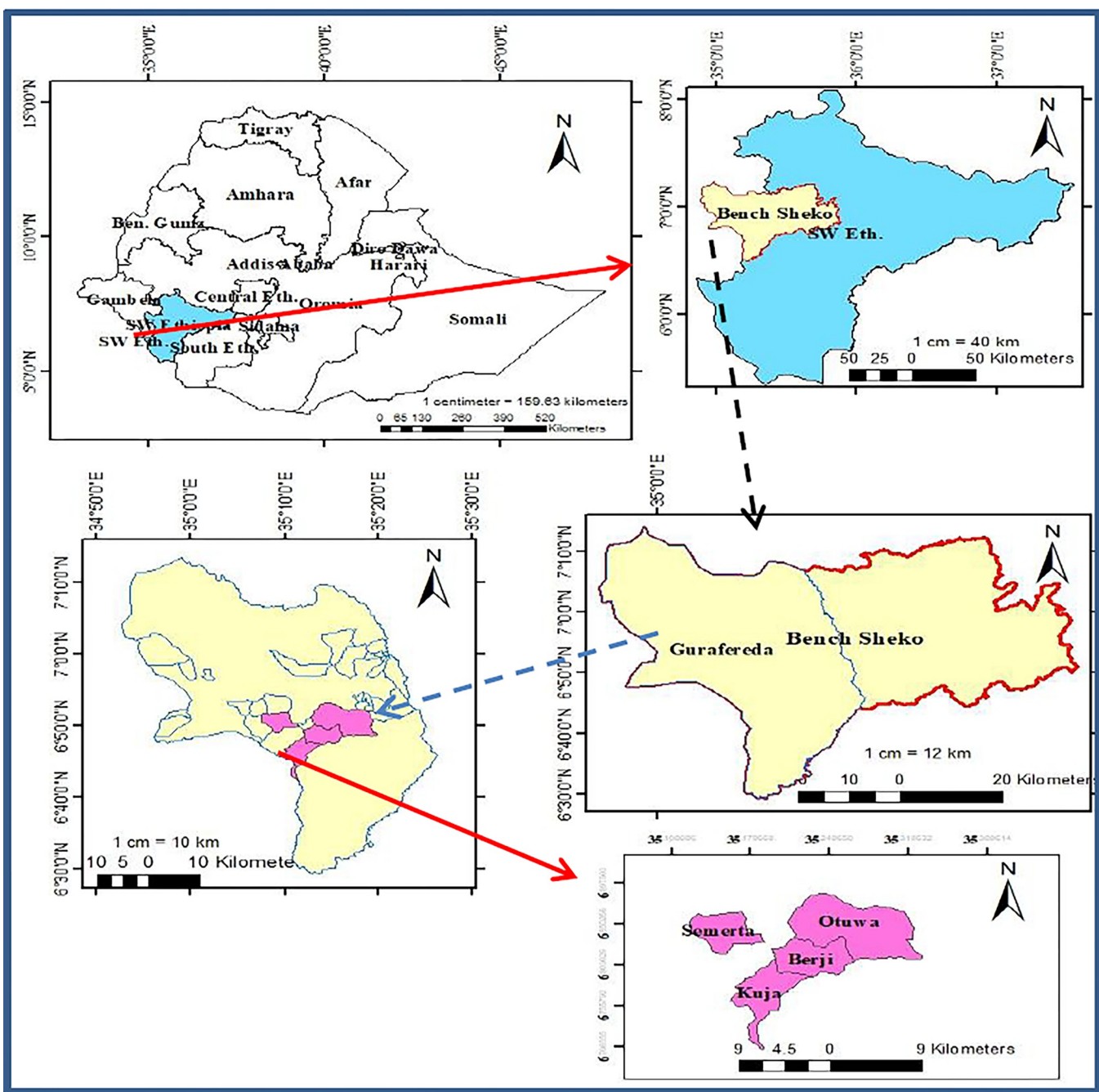

**Fig 1. Map of the study area (Source: ARCGIS, 2024).**

data considered to be important were collected from different sources including unpublished administrative documents of different organizations and published sources.

### 2.4 Method of data analysis

To realize the objectives of the study, two analytical tools were employed, namely, descriptive and econometric models.

## 2.5 Descriptive statistics

The sample respondents' demographic and socio-economic conditions were analyzed using descriptive statistics like mean, standard deviations, frequency, and percentage by using Statistical Package for Social Science (SPSS).

## 2.6 Multiple linear regression model

To analyze determinants of market supply of rice multiple linear regression model was used. The model was found to be appropriate because all the sample respondents supplied positive amount of rice to the market. The multiple linear regression model was specified as follows.

To analyze determinants of market supply of rice multiple linear regression model was used. The multiple linear regression model employed was specified as shown below.

$$Y_i = \beta_0 + \beta_1 SEX + \beta_2 AGE + \beta_3 HHS + \beta_4 FASIZE + \beta_5 RPRODCD + \beta_6 CREDT + \beta_7 INCOM + \beta_8 MARKET + \beta_9 EXTEN + \beta_{10} FERTLZR + \beta_{11} SEED + \beta_{12} EDUCTN + \beta_{13} OXEN + \mu$$

Where, $Y_i$ is, market supply of paddy rice, $\beta_0$ is intercept, $\beta_1$-$\beta_{13}$ denotes, coefficients to be estimated in relation to associated explanatory variables, SEX is sex of household heads, AGE is age of respondents, HHS is household size, FASIZE is total farm size, RPRODCD is rice quantity produced in a year, CREDT is credit used by a households, INCOM is annual income, MARKET is distance from nearest market, EXTEN is extension contact, FERTLZR is use of inorganic fertilizer, SEED is use of improved rice varieties, EDUCTN is education level, OXEN is, number of oxen owned and μ is Random error term. The summary of definitions of variables and working hypothesis shown below (Table 1).

# 3. Results and discussion

## 3.1 Descriptive analysis

The result of descriptive analysis discussed in this section. From the respondents 86.6% were married, 7.5% were divorced and 6.7% were widowed. Most of the respondents (87.6%) were male-headed and 12.6% were female-headed households (Table 2). The average age of the respondents was 46.9 years with a minimum of 26 years and a maximum of 74 years. Farm

**Table 1. Summary of definitions of variables and working hypothesis.**

| Variables | Descriptions | Types | Expected sign |
| --- | --- | --- | --- |
| SEX | Sex (male = 1 and female = o) | Dummy | + |
| AGE | Age of household in years | Continuous | + |
| EDUCTN | Education level (in years) | Continuous | + |
| HHSIZE | House hold size | Continuous | - |
| FARMICOME | Farm income (ETB in thousands) | Continuous | + |
| FARMSZ | Farm Size (in hectares) | Continuous | + |
| OXEN | Number of oxen owned | Continuous | + |
| RICEPRD | Quantity of rice produced | Continuous | + |
| MAKTDTC | Market distance (in Km) | Continuous | - |
| EXTN | Extension contacts (yes = 1 and No = 0) | Dummy | + |
| CREDIT | Credit use (yes = 1 and No = 0) | Dummy | + |
| FERTLZR | Use of inorganic fertilizers (yes = 1 and No = 0) | Dummy | + |
| IMPVDVATY | Use of improved varieties (yes = 1 and No = 0) | Dummy | + |

**Table 2. Descriptive analysis of continuous variables.**

| Parameter | Minimum | Maximum | Mean | Std. Deviation |
|---|---|---|---|---|
| Age | 26.00 | 74.00 | 46.91 | 10.351 |
| Education level (in years) | 0.00 | 9.00 | 1.41 | 2.536 |
| Household size | 2.00 | 12.00 | 6.13 | 2.017 |
| Annual income (ETB in thousands) | 2.00 | 60.00 | 20.983 | 11.94 |
| Total farm size (hectares) | 0.25 | 6.00 | 2.2941 | 1.18 |
| Rice quantity produced (in tons) | 0.40 | 8.30 | 2.80 | 1.60 |
| Rice sold (in tons) | 0.00 | 7.50 | 1.91 | 1.48 |
| Oxen owned | 0.00 | 4.00 | 1.41 | 0.942 |
| Market distance (km) | 3.00 | 18.00 | 9.36 | 3.82 |

activity was the basis for most farmers' livelihood as it was seen as a major source of household income and life support services. The average annual income of the respondents is 20,983 birrs (Ethiopian currency), which was mainly derived from agriculture. Farmers on average hold land size of 2.14 hectares. The mean yearly production of rice at the household level reaches one hectare with a mean production of nearly 2.8 tons per hectare. Farmers sold on average 70% of their produced paddy rice to grain traders at the village level or district level; while value addition activities, like selling milled rice were not practiced by farmers (Table 2).

## 3.2 Determinants of market supply of rice

The linear regression model results indicated in Table 3 below show that the supply of rice to the market was determined by the interaction of different demographic, economic, and institutional factors. Before conducting regression, the correlation of dependent variables and independent variables was checked and those variables satisfying the correlation conditions were included for regression analysis. The presence of multicollinearity among explanatory

**Table 3. OLS regression results for sales volume of rice.**

| Variables | Coefficient | Std. Error. | t-ratio |
|---|---|---|---|
| Sex of household head | -0.015 | 0.102 | -0.15 |
| Age of household head | -0.004 | 0.004 | -0.91 |
| Education level | 0.001 | 0.013 | 0.72 |
| Household size | 0.031 | 0.021 | 1.45 |
| Farm size | 0.097 | 0.042 | 2.27** |
| Rice quantity produced | 0.063 | 0.004 | 15.33*** |
| Fertilizer application | 0.063 | 0.065 | 0.95 |
| Improved variety use | 0.067 | 0.094 | 0.70 |
| Distance to the market | -0.001 | 0.0088 | -0.21 |
| Extension contacts | -0.053 | 0.081 | -0.66 |
| Credit use | 0.168 | 0.079 | 2.13** |
| Farm income | 0.021 | 0.004 | 4.73*** |
| Oxen number owned | 0.150 | 0.057 | 2.66*** |
| Constant | -0.764 | 0.21 | -3.62 |

Number of observations = 119 R-squared = 0.954.

Root MSE = 3.385 Adj R-squared = 0.9484.

$F_{(13, 106)}$ = 167.67 Prob.> F = 0.0000.

variables was checked and there was no serious problem. From the total 12 explanatory variables included in the econometric model, five variables were found to have a significant influence on the market supply of rice. These are household size, the quantity of rice produced, total farm size, total farm income, and credit use.

Farm size relation with sales volume was positive and influenced the marketable supply of rice positively at a 5% level of significance. This shows that if the farm size of a household increases by a unit or simply by a hectare, the supply of rice to the market increases by 0.097 tons holding other things constant. This can be explained by the fact that farm households that have larger land sizes produce rice in larger amounts and have the probability to supply more. In addition, through diversification, they produce other crops like maize, sorghum, and others that can support household consumption needs, and since rice has a better market price than other crops, more rice is supplied to the market. The land is an important factor in production and the larger the size of productive land the producer owns, hence the higher the production levels are likely to be due enabling a household to produce a market surplus and be gifted to sell a substantial amount of produce [19].

Oxen owned was hypothesized to have a positive influence on the dependent variable and in the regression model it has shown a positive sign on the coefficient and the variable influenced the market supply of rice positively at a 1% level of significance. This shows that if the number of oxen owned by a farmer increases by a unit, the supply of rice to the market increases by 0.15 tons holding other things constant. Since land preparation in the district is merely done by using oxen, having more of it enables producers to prepare land on time and effectively to produce rice in larger plots and supply more to the market. The finding is similar with previous study by [15].

The quantity of rice produced affected sales volume in a positive relation and influenced the market supply of rice at a 1% level of significance. This shows that if the quantity produced increases by a unit, the supply of rice to the market increases by 0.063 tons holding other things constant. Rice in the district was produced in a better market-oriented way than other crops. It can be explained by a farmer who produces more would probably supply more to the market. In line with this finding, [20] also showed higher yield increases the farmer's likelihood to participate in the market because the surplus above their household consumption needs makes more supply to the market. The quantity of rice produced is positively related to the intensity of market participation as the quantity produced is critical for semi-commercial farmers who first have to produce for home consumption and only sell surplus [19].

Credit use influenced the market supply of rice positively and significantly at a 5% level of significance. This shows that if households participated in credit, the supply of rice to the market increased by 0.16 tons holding other things constant. This might be due to the reason that rice producer households with labor and financial shortages need to use hired labor for land preparation, weeding, and harvesting, as a result, they utilize credit for paying labor and fertilizer costs for rice production in larger amounts. In line with this, other researchers also showed that access to credit enables producers to increase the amount of inputs and other inputs (fertilizer, seed, plows) which in turn boosts output produced and surplus for the market [19].

Total farm income influenced the market supply of rice positively at a 1% level of significance. This shows that if households' annual income (in thousands), increases by a unit, the supply of rice to the market increases by 0.02 tons holding other things constant. It might be due to a farmer with a better annual income having a better opportunity to produce rice in larger size by employing all necessary production inputs and supplying more rice to the market than those with lower income.

**Table 4. Constraints related with rice production.**

| What are constraints in rice cultivation? | Yes (N = 119) | Percent | St. Deviation |
|---|---|---|---|
| Lack of market and market information | 22 | 18.6 | 0.390 |
| Fertilizer application &price | 90 | 75.6 | 0.431 |
| Improved seed supply shortage | 60 | 50.4 | 0.502 |
| Land shortage | 28 | 23.5 | 0.426 |
| Disease and insect pest occurrence | 48 | 40.3 | 0.493 |
| Transport shortage | 15 | 14 | 0.409 |
| Institutional support problem | 87 | 73.1 | 0.445 |
| Extension service provision | 64 | 53.8 | 0.501 |
| Price setting & scaling problem | 70 | 58.8 | 0.494 |
| Crop management | 69 | 58 | 0.482 |

### 3.3 Constraints related to rice production in the district

Constraints related with rice production was analyzed based on data collected from rice producer households. Major constraints identified were discussed below as shown in the Table 4 below.

### 3.4 Inorganic fertilizer application

Most of the respondent farmers mentioned that the specific time of fertilizer application, type and amount as well as way of application was not clearly known and, in the event, that unless these conditions were fixed, they believe that fertilizer application contributed no significant yield difference. Due to this, some farmers were hesitant to apply inorganic fertilizer on rice as they believed that there was no positive significant change in the yield. Even though some farmers accept extension agents' advice on the provision and application of inorganic fertilizer to rice fields, some of them use it inappropriately seemingly showing unnecessary resource wastage. According to [21], the efficiency of nitrogen applied to rice crops depends on the type of fertilizer, timing of application, seasonal trends, and others related factors. However, it needs critical care during decision making in relation to rate, application date and crop growth stage. However, in Ethiopia, there is a gap in conducting in-depth research studies to improve the grain yield except few studies done on different N rates on a few rice cultivars. According to [22] rice yield in the district is low due to lack of area-specific recommendations on fertilizer rate.

### 3.5 Weak crop management practice

Rice needs higher management practice especially, at the time of weeding than other major crops grown in the area. For farmers who produce it intensively, weeding becomes a difficult task as weeding is done by hand. Despite most of the farmers in the area using herbicide for weed control, its ineffectiveness on grassy weed and Mimosa invisa weed made hand weeding the only option with intensive labor force employment. [23] indicated that the rice yield losses due to weed interference in the district can reach 68%. In addition, rainfall shortage during a critical rice growing period was affecting yield highly and this shows that the issue needs further soil and water management practices and variety selection for moisture stress tolerance.

### 3.6 Weak institutional support

Problems related to security problems sometimes cause conflict, extension service provision, weak research-extension-farmer linkage, administrative issues related to land ownership

certification, and weak monitoring of traders and markets by concerned government organizations were problems farmers pointed out. Strong institutional support along the value chain of rice contributes to effective value chain development for the commodity. Farmers' smooth participation along different value chain actors like research organizations, cooperatives, extension, and other development associations including NGOs is necessary to improve farmers' rice production experience [24]. However weak institutional support in the district was mentioned as one of the major challenges in the district.

### 3.7 Improved seed supply shortage

Most of the farmers were using well-known local rice varieties and little improved varieties. However, farmers need high-yielding improved varieties critically. According to the farmers' response, the existing newly supplied rice varieties, especially, NERICA-4 have a threshing problem after harvesting, unlike the local one which is easily threshed immediately after harvest. In addition to this, NERICA 4 has a short height characteristic and it is incompatible with the local farming system. As a result, compared to local varieties the improved varieties were not adopted by most of the producers yet. The need for other promising new varieties is continued as it holds a high-value question for researchers to provide varieties to increase productivity sustainably in the district. [11] revealed the contribution of access to improved rice varieties via that adoption helps to participate in rice markets and that in turn had a higher significant impact on welfare indicators; such as consumption expenditure, rice income, average yield, and access to credit.

### 3.8 Lack of effective extension service support

In the area, the extension services are limited only to fertilizer supply, rather there was a weak focus on agronomic management practices, crop protection, and marketing services. Due to this reason, some farmers face a knowledge gap in agronomic crop management practices. Crop and animal production with frequent and effective monitoring and support were some of the major provisions that farmers need from extension service providers. Extension service based on farmers' participatory research and training programs contributes to increased dissemination of information and demonstration and popularize new rice technologies; showing the need to consider farmers' preferences for new varieties and others through research and demonstrations at farmers' fields to enhance and promote the adoption [25].

### 3.9 Disease and pest problem

According to the sample farmers' response, one of the most important related rice production constraints was disease. Farmers identified it locally known as "MICHI" which causes a high amount of yield loss. The symptoms of the disease that respondents mentioned were early related to that of rice brown spot. According to [26], severe outbreaks of rice disease and pests caused a significant number of yield losses. Fungal diseases, such as sheath rot, sheath brown rot, and blat are the most common diseases. The low productivity of rice in Ethiopia is attributed to several factors, important of which are seed-borne diseases, causing 50–80 percent yield losses. Among these, blast disease was reported in almost 85 rice-growing countries of the world. According to [27] fungi, viruses, bacteria, insects, and nematodes are causing problems in rice crop and have potential to affect production and productivity if appropriate solution not provided timely.

### 3.10 Price setting & scaling problem

Most farm households use rice as a cash crop and they cover their cash needs for fertilizer costs, oxen buying, schooling costs, and others by selling at the time they need. Respondents

said that some traders communicate with each other and make the rice price lower at the time when they bring the rice to the market and that affects the bargaining power of the farmers. This condition makes farmers sell their produce at a lower price than bringing the product back to the home if cash needs are urgent. The market related problems faced by farmers in marketing their output is due to the lack of clear information about the price level of product in the market as the price is easily manipulated by traders [28]. As the rice production and marketing is considered as new phenomena compared to other cereal commodities, rice market is not competitive. The finding by [29] reveled that rice the structure was strongly oligopolistic by which the market was governed by few wholesaler traders.

### 3.11 Shortage of threshing and milling machine

Due to the unavailability and high cost of milling machines, all the farmers are selling their rice in the form of paddy and this incapacitates farmers to benefit from better prices by adding value to their product. The difficulty of threshing for the NERICA-4 variety as a problem was mentioned by the producers which was one of the main reasons for the lower adoption of the variety. The rice production is done by hand or with traditional tools as only 2% of farmers have access to mechanization. This has contribution for low quality rice supply as rice harvesting and threshing are done manually using a serrated sickle and animal trembling [30].

### 3.12 Lack of market

The absence of better competition among traders in the district narrowed the alternative possibility of getting better prices at better markets for rice output. However, some produces mentioned the situation is better compared to what was happening several years back.

## 4. Conclusions

In the study area, most of the smallholder farmers were producing rice in a better market-oriented way than other crops grown in the area and it created better employment opportunities for most farm households. Rice was being produced as a cash crop by most farmers as they supplied more of their product (more than 68%) to the market to make money for farming and household consumption expenditures. Determinants of the market supply of rice were total farm size, total household size, credit use, total farm income, and quantity of rice produced.

Major problems related to rice production in the area were a lack of improved rice production technologies, the requirement of heavy crop management especially weeding, fertilizer supply and high prices, problems related to security problems sometimes causing conflict, weak extension service provision, weak research-extension-farmer linkage, weak monitoring of traders and market by concerned government organization. The rice crop is a significant contributor and the same will be in the future as the crop has considerable potential to improve the livelihood of farm households and communities. Thus, attention should be given to rice production and marketing-related constraints.

## 5. Policy implications

Rice is a major crop that can be considered as a base for the livelihoods of households in a district, improved rice technologies are necessarily needed to benefit smallholder farmers. Thus, attention should be given to rice production-associated constraints. Introduction, evaluation, and generation of improved high-yielding rice varieties and other demand-driven rice

technologies for producers should be provided by concerned government and nongovernmental organizations via strong research extension and farmer linkage.

Since rice production requires heavy management practices and credit use has shown a positive relation with marketed rice supply, rural finance outreach and access to every smallholder farmer need to get attention to increase rice production and market supply. Small family-sized households with financial shortages face labor shortages for rice production, especially at the weeding time, and consequently, their marketed rice supply becomes lower and they need to be supported by improved crop management technologies especially, weed control systems.

Most extension agents joining the rice producer community after graduation from college have weak practical experience in rice production. Therefore, building the capacity of extension agents through effective training should get attention to increase production and productivity including improvement in the quality of rice products through better post-harvest handling and processing.

## Supporting information

**S1 File.**
(DTA)

## Acknowledgments

The author would like to acknowledge Southern Agricultural Research Institute for the overall facilities provided during the execution of this study. Particular gratitude is extended to all the respondent who took part in the research data collection by providing insightful data.

## Author Contributions

**Conceptualization:** Ejigu Mulatu.

**Data curation:** Ejigu Mulatu.

**Formal analysis:** Ejigu Mulatu.

**Investigation:** Ejigu Mulatu.

**Methodology:** Ejigu Mulatu.

**Project administration:** Ejigu Mulatu.

**Resources:** Ejigu Mulatu.

**Software:** Ejigu Mulatu.

**Supervision:** Ejigu Mulatu.

**Validation:** Ejigu Mulatu.

**Visualization:** Ejigu Mulatu.

**Writing – original draft:** Ejigu Mulatu.

**Writing – review & editing:** Ejigu Mulatu.

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
