## [Decision Letter · Decision Letter 0]

29 May 2024

PONE-D-24-11303Factors Affecting Rice Production and Market Supply: the case of Guraferda district, Bench Maji zone in Southern EthiopiaPLOS ONE

Dear Dr. Mulatu,

Thank you for submitting your manuscript to PLOS ONE. After careful consideration, we feel that it has merit but does not fully meet PLOS ONE’s publication criteria as it currently stands. Therefore, we invite you to submit a revised version of the manuscript that addresses the points raised during the review process.

We look forward to receiving your revised manuscript.

Kind regards,

Meraj Alam Ansari

Academic Editor

PLOS ONE

3. In the online submission form, you indicated that [Data will be available upon the request].

Additional Editor Comments:

Comments

Change the title its looking local study, it should be region specific title "Determinants of Rice Production and Market Supply: A Study of Bench Maji Zone, Ethiopia"

Abstract should be supported with quantified data of your results.

Define all the determinants taken in this study and support with quantified data as well as references. Introduction is too long and vague statements must be discarded from introduction. Focused introduction should be rewritten.

Study area map should be included

Conclusion looks like a thesis finding. It should be very sound and brief.

Policy implication should be added separately.

Check your references properly, excess reference should be deleted and missing references in text should be added in the list.

Punctuation and grammatical error should be rectified through out the manuscript.

Based on the reviewwers comments the article should be accepted after Major Revision. All the comments should properly address.

Reviewers' comments:

Reviewer's Responses to Questions

**Comments to the Author**

1. Is the manuscript technically sound, and do the data support the conclusions?

Reviewer #1: Yes

Reviewer #2: Yes

2. Has the statistical analysis been performed appropriately and rigorously? 

Reviewer #1: Yes

Reviewer #2: Yes

3. Have the authors made all data underlying the findings in their manuscript fully available?

Reviewer #1: Yes

Reviewer #2: Yes

4. Is the manuscript presented in an intelligible fashion and written in standard English?

Reviewer #1: Yes

Reviewer #2: Yes

5. Review Comments to the Author

Reviewer #1: After reviewing the manuscript, following are my comments for the author:

1. The title should clearly reflect the specific focus of the study. Consider revising to something more precise, such as "Factors Influencing Smallholder Farmers' Market Supply of Rice in Guraferda District, Ethiopia."

2. Language and Grammar: Revise for grammatical accuracy and readability.

3. The results can be improved by adding figures.

4. Comparing and contrasting with similar studies would add depth to the discussion.

Reviewer #2: The paper is suitable for publication; however, the author should provide a brief development status of rice production and the relevance of market supply for sustainable rice production and household economy.

6. PLOS authors have the option to publish the peer review history of their article (what does this mean?). If published, this will include your full peer review and any attached files.

Reviewer #1: No

Reviewer #2: **Yes: **Dr. Cornel Anyisile Kibona

---

## [Author Response · Author response to Decision Letter 0]

13 Jul 2024

Dear,

Meraj Alam Ansari

Thank you for giving me the opportunity to submit a revised manuscript titled: Factors Affecting Rice Production and Market Supply: the case of Guraferda district, Bench Maji zone in Southern Ethiopia to PLOS ONE Journal. 

I appreciate your time and effort that journal editor and the reviewers have dedicated to providing your comment to modify the manuscript. I am really grateful to the editor and reviewers for their insightful comments on improving my paper and make sound. I have been able to incorporate all necessary comments and suggestions provided by the editor and reviewers. I have made the changes within the manuscript with the track change and submitted with also the revised manuscript for publication. Herewith I have provided response to all the relevant reviewers’ comments.

Comments from Editor

Comment1: Change the title its looking local study, it should be region specific title Determinants of Rice Production and Market Supply: A Study of Bench Maji Zone, Ethiopia"

Response: Accepted and modified the title accordingly

Comment 2: Abstract should be supported with quantified data of your results.

Response: Accepted and modified with thanks

Comment 3: Define all the determinants taken in this study and support with quantified data as well as references. 

Response: All available reviews were done and all determinants defined 

Comment 4: Introduction is too long and vague statements must be discarded from introduction. Focused introduction should be rewritten.

Response: This comment has been taken as a very critical and all necessary adjustments were made and introduction part made more condensed and shortened. 

Comment 5: Study area map should be included

Response: Accepted and map is included 

Comment 6: Conclusion looks like a thesis finding. It should be very sound and brief.

Policy implication should be added separately.

Response: Accepted and made brief

Comment 7: Check your references properly, excess reference should be deleted and missing references in text should be added in the list.

Response: Reference part has been double checked and necessary adjustments done

Comment 8: Punctuation and grammatical error should be rectified throughout the manuscript.

Response: Checked thoroughly the paper and modified 

Comments from the reviewer 1: The manuscript must describe a technically sound piece of scientific research with data that supports the conclusions. Experiments must have been conducted rigorously, with appropriate controls, replication, and sample sizes. The conclusions must be drawn appropriately based on the data presented.

Response: As the study was based on household survey, there is no experiment conducted and conclusion drawn based on the data presented. 

Comments from the reviewer 4: Any typographical or grammatical errors should be corrected at revision, so please note any specific errors here.

Response: Accepted and all necessary modifications made

Comments from reviewer 5: 

Comment 1. The title should clearly reflect the specific focus of the study. Consider revising to something more precise, such as "Factors Influencing Smallholder Farmers' Market Supply of Rice in Guraferda District, Ethiopia."

Response: As the study was mainly focused on market supply and production, the author wanted to mention the title as the editor commented. 

Comment 2. Language and Grammar: Revise for grammatical accuracy and readability.

Response: Accepted and all necessary modifications made

Comment 3. The results can be improved by adding figures.

Response: depending on the type of study, could not find appropriate figure to add. Only map of the study area added on methodology part.

Comment 4. Comparing and contrasting with similar studies would add depth to the discussion.

Response: Accepted and all necessary modifications made

Sincerely, 

Ejigu Mulatu

---

## [Editor Report · Decision Letter 1]

18 Jul 2024

PONE-D-24-11303R1Determinants of Rice Production and Market Supply: A Study of Bench Sheko Zone in EthiopiaPLOS ONE

Dear Dr. Mulatu,

Thank you for submitting your manuscript to PLOS ONE. After careful consideration, we feel that it has merit but does not fully meet PLOS ONE’s publication criteria as it currently stands. Therefore, we invite you to submit a revised version of the manuscript that addresses the points raised during the review process.

We look forward to receiving your revised manuscript.

Kind regards,

Meraj Alam Ansari

Academic Editor

PLOS ONE

Journal Requirements:

Additional Editor Comments:

Dear Author

The manuscript is substantially revised by the Authours. It may accepted after minor correction as suggested below.

1. Introduction section may focussed with inclusion of Keywords, gap, novelty of study, robust hypothesis etc.

2. Refrences must be arranged in the format of PLOS One. In refernce No. 14,22, 27, 28 year of publication is missing.

Thankyou

---

## [Author Response · Author response to Decision Letter 1]

24 Jul 2024

All the comments were well received and incorporated. The comments were very critical to improve the manuscript. 

Comment on reference 

Comment1: Follow journal reference style and make complete and correct. Comment on retracted articles 

Response: Accepted and modified all the references used. All relevant changes were made. I have not used retracted articles.

Editor Comments:

1.Introduction section may focused with inclusion of Keywords, gap, novelty of study, robust hypothesis etc.

Response: The comment is accepted and all necessary improvements were made. 

2. References must be arranged in the format of PLOS One. In refernce No. 14,22, 27, 28 year of publication is missing.

Response: The comment is accepted and all necessary improvements were made. Major revision in reference section made and hopefully fits the journal requirement. Thank you for this very important and critical comment.

Thank you very much.

---

## [Editor Report · Decision Letter 2]

8 Aug 2024

Determinants of Rice Production and Market Supply: A Study of Bench Sheko Zone in Ethiopia

PONE-D-24-11303R2

Dear Dr. Mulatu

We’re pleased to inform you that your manuscript has been judged scientifically suitable for publication and will be formally accepted for publication once it meets all outstanding technical requirements.

Kind regards,

Meraj Alam Ansari

Academic Editor

PLOS ONE
---

## [Editor Report · Acceptance letter]

27 Aug 2024

PONE-D-24-11303R2 

PLOS ONE

Dear Dr. Mulatu, 

I'm pleased to inform you that your manuscript has been deemed suitable for publication in PLOS ONE. Congratulations! Your manuscript is now being handed over to our production team.

Kind regards, 

on behalf of

Dr. Meraj Alam Ansari 

Academic Editor

PLOS ONE